# Complementary Photostimulation of Seeds and Plants as an Effective Tool for Increasing Crop Productivity and Quality in Light of New Challenges Facing Agriculture in the 21st Century—A Case Study

**DOI:** 10.3390/plants11131649

**Published:** 2022-06-22

**Authors:** Agnieszka Klimek-Kopyra, Tomasz Czech

**Affiliations:** 1Department of Agroecology and Plant Production, University of Agriculture in Kraków, Al. Mickiewicza 21, 31-120 Kraków, Poland; 2Department of Agricultural and Environmental Chemistry, University of Agriculture in Kraków, Al. Mickiewicza 21, 31-120 Kraków, Poland; tomasz.czech@urk.edu.pl

**Keywords:** soybean, yield optimization, chemical composition

## Abstract

Climate change has prompted the search for new methods for improving agricultural practices for legume crops. The aim of the study was to test an innovative method of complementary photostimulation of seeds and plants aimed to improve the quantitative and qualitative features of soybean (*Glycine hispida* L. (Merr.)) yield. Complementary photostimulation of plants was shown to positively affect the yield and chemical composition of soybeans, significantly increasing the content of protein and fat in seeds of the Merlin cultivar. Significant positive effects compared to the control were obtained following irradiation of seeds and plants for 3 s (the shorter of the analyzed exposure times). The results clearly indicate the need to improve the proposed new HUGO (High Utility for Optimal Growth) technology to optimize soybean yield.

## 1. Introduction

The increase in the global population since the start of the 20th century has led to significant degradation of agroecosystems. To minimize progressive degradation of the agricultural environment, the concept of sustainable development was created and introduced to the agricultural sector, with the aim of balancing the three overriding goals of crop production: ecological, economic and social [1,2]. Lewandowksi et al. [3] defined sustainable agriculture as the management and use of the agricultural ecosystem in a manner that preserves its biodiversity, productivity, regeneration, vitality and capacity to perform essential ecological, economic and social functions at the local and global level without harming other ecosystems. Despite the fact that sustainable development has been known for nearly 30 years, it is only recently begun to be appreciated by large agricultural producers, who on the one hand have increased environmental awareness and want to develop farms in a sustainable manner, and on the other hand are obliged to take greater care of the environment under public pressure and new EU Common Agricultural Policy [4,5]. Tangible ecological, economic and social indicators show that agricultural producers face the serious challenge of increasing food production in a sustainable manner. It is becoming increasingly clear that technological innovations associated with soil cultivation and application of plant protection products and fertilizers that enable a rapid increase in food productivity are not the best. Therefore, the challenges associated with future sustainable crop production will involve minimizing degradation of the environment caused by overexploitation of natural resources, improving environmental safety by restricting the use of plant protection products and mineral fertilizers, increasing biodiversity and popularizing biostimulators. It is believed that in the next few years, plant biostimulators, including both biological and physical methods, can be expected to not only significantly support sustainable crop production systems in agricultural agroecosystems less affected by chemicals, but also to become an element of a modern system of sustainable agriculture based on new technologies [6,7].

In the near future, the stimulation of plants by biological methods using diverse biostimulators will be an essential ‘biological tool’, enhancing nutrient use efficiency and nutrient acquisition by plants, but also a tool for decreasing synthetic fertilizer consumption. However, the presented approach still requires the interdisciplinary knowledge of scientists working in multiple fields. Du Jardin [8] showed that combining knowledge from various fields of science (molecular biology, biochemistry and physiology) will open up new perspectives for the creation of effective biostimulators for optimizing crop production. Yan et al. [9] suggested that in the near future interdisciplinary research on bacterial communities associated with AMF (arbuscular mycorrhizal fungi) should be adopted as an integrated tool for identifying the best combinations of bacterial communities capable of increasing the efficiency of nutrient utilization and the resistance of crop plants and of enhancing nutraceutical compounds among species of crop plants. Dłużniewska et al. [10] presented interdisciplinary results of studies, which combined two different methods of biostimulation, i.e., biological stimulation through the application of a mycorrhizal inoculum and physical stimulation through the application of laser light to seeds. This method has unlimited potential due to the positive synergistic effect obtained when it is used in combination with other methods of seed biostimulation, such as biological methods [10,11]. Dłużniewska et al. [10] showed that the health of young soybean plants was improved. The stimulation of the soybean seeds and AMF inoculum reduced the incidence of Septoria brown spot but only slightly reduced the occurrence of Fusarium root rot. The use of the AMF inoculum (irradiated or not) as a factor improving the condition of plants had a negative effect on plant productivity. A slight decrease in seed weight per plant and pod number per plant was observed. For plant health, a stimulation of seeds coated with AMF inoculum is recommended as a method of seed enhancement.

Biostimulation of seeds by physical methods still seems to be unknown, despite the continuous technological progress in the optimization of crop fertilization. The method of biostimulation of seeds with laser light was described in the 1960s, but due to technological barriers it has not been used in practice [6]. Currently, new possibilities for the practical use of biostimulation methods are perceived. Biostimulation of seeds with laser light increases seed germination efficiency and accelerates initial growth and development. 

An innovative method of using low-level lasers to accelerate the growth of plant biomass, leading to an increase in plant production, was described by Klimek-Kopyra et al. [6]. The authors explained that the effect of biostimulation on plant biology. Directly after photostimulations of plant tissues, the absorption of quanta of laser light with high energy density by enzymes, e.g., mitochondrial enzymes, and nucleic acids is observed. A visible effect of biostimulation occurs when the laser light acts on biological material and is accompanied by a local increase in temperature (0.5 °C), and the changes taking place at the cellular level, caused by the absorption of radiation, are not the tissues’ response to stress or to their destruction [6,12]. Klimek-Kopyra et al. [6] noted, followed by Budagovsky et al. [13], that laser irradiation can contribute to the photoinduction of processes through its effect on the microstructures of biological material. Stimulation can cause changes in cellular metabolism by accumulating energy in ATP and through the transport of electrons in the respiratory chain. The wrong parameters or excessively high irradiation intensity can lead to photodestruction of chlorophyll in plants. This is the reason why further studies were undertaken in field conditions.

### 1.1. Effect of Laser Photostimulation on Plant’s Physiology

Pre-sowing laser photostimulation of seeds attracts the keen interest of the scientific community because of its potential to enhance seed germination, seedling growth, physiological, biochemical and yield attributes of plants, cereal crops and vegetables [6,14,15,16,17].

Asghar et al. [14] assessed the laser light (HeNe—wavelength 632 nm and density power of 1 mW/cm^2^) effect on biochemical, enzyme activities and chlorophyll contents in soybean seeds and seedlings during the early growth stages. Authors proved that laser pre-sowing seed treatments have potential to enhance soybean physiological and biochemical parameters, e.g., chlorophyll contents (Chl “a” “b” and total chlorophyll contents) and metabolically important enzymes (degrade stored food and scavenge reactive oxygen species), compared to the control (untreated seeds). However, future studies should be focused on growth characteristics at later stages and yield attributes. Perveen et al. [15] evaluated pre-sowing laser treatment of sunflower (*Helianthus annuus*) seeds on various biochemical, physiological, growth and yield parameters under greenhouse conditions. The authors proved that the physiological attributes such as photosynthetic rate (*A*), transpiration rate (*E*), intrinsic CO_2_ concentration (*C*_i_), stomatal conductance (*g*_s_), chlorophyll *a* and *b* contents, relative membrane permeability and leaf water (*ψ*_w_), osmotic (*ψ*_s_) and turgor (*ψ*_p_) potentials, relative water contents and leaf area increased significantly compared to the control due to He-Ne treatment of seeds. The activities of superoxide dismutase, peroxidase and catalases and contents of total soluble proteins, malondialdehyde, proline and leaf total phenolic also increased due to laser treatment. Significant increase in growth parameters of sunflower such as shoot fresh and dry masses, root fresh and dry masses, root and shoot lengths, number of leaves per plant and stem diameter has also been observed. The contents of K, Ca and Mg in shoot and root were also increased and an overall increase of up to 28.12% was observed due to laser treatment.

Samiya et al. [16] compared the effects of different diode laser irradiation (green light vs. red light) on wheat (*Triticum aestivum*) seed germination, early growth and biochemical parameters. The authors proved that laser irradiation treatment had significant effects on the biochemical parameters such as superoxide dismutase (SOD), peroxidase (POD), catalase (CAT) and protein concentrations. The red laser treatment showed significant effects for germination and growth parameters such as the number of roots, number of shoots, germination percentage, dry and fresh weight of shoot length, dry and fresh weight of root length, protein percentage and POD, while the green laser treatment showed significant effects on parameters such as root length, shoot length, SOD and CAT. The above findings [14,15,16] reveal positive effects of pre-sowing laser photostimulation on plant physiology, especially on morphological and biochemical parameters. 

In view of the new challenges [18] related to the need to increase the productivity of plants and improve their utility values in order to feed 9 billion people by 2050 and beyond, there is a need to develop the existing pre-sowing laser stimulation method used in agriculture and present the new approach, which will be inspirited by scientists from different academic fields.

### 1.2. The Concept of HUGO Technology

HUGO (High Utility for Optimal Growth) technology involves complementary photostimulation of seeds in combination with the irradiation of small plants in field conditions in order to improve their productivity and health. Complementary HUGO technology consists of a two-step photostimulation (Figure 1). In step one, biostimulation of seeds before sowing takes place in controlled conditions or directly before sowing using a seeder machine suitably modified for the irradiation of seeds. Step two involves the biostimulation of juvenile plants in field conditions using a mobile platform with a modular structure equipped with a set of low-power lasers. The means of irradiation of seeds and plants will depend on the species. HUGO technology is mainly intended for species which are grown at wide spacing, have limited, slow growth in the initial stages of vegetative development and are vulnerable to fungal diseases. 

The aim of the study was to assess the effectiveness of complementary photostimulation of seeds and plants to optimize agricultural practices for legume species based on the example of the Merlin cultivar of soybean. 

## 2. Material and Methods

### 2.1. Case Study—Experimental Design

A single-factor experiment was performed in field conditions. The microplot experiment was set up on brown soil with optimal nutrient content (pHkcl = 6.8, available forms of phosphorus in soil 26.2 mg 100 g^−1^, whereas potassium 16.9 mg 100 g^−1^, content of organic matter 2.1% average). The field conditions were selected based on soil analysis in order to eliminate nutrient limitation. Certified, uniform soybean seeds of the Merlin cultivar, from the company Saatbau were used for testing. Soybean of the Merlin cultivar was sown in the first 10 days of May and harvested in the last 10 days of September. The area of each microplot was 7 m^2^. The width between rows was 25 cm. The seed density was 50 seeds per square meter.

Six different biostimulation combinations and a control treatment were compared, according to the experimental design shown in Table 1: K1—3 × 3 s seed stimulation; K2—3 × 9 s seed stimulation; K3—3 × 3 s plant stimulation; K4—3 × 9 s plant stimulation; K5—3 × 3 s seed and plant stimulation; K6—3 × 9 s seed and plant stimulation; and K7—control. Biostimulation was carried out in 2 steps. The first step involved the biostimulation of soybean seeds in controlled conditions before sowing (K1, K2, K5 and K6). A stationary device (European Patent) was used to stimulate the seeds with a low-power diode red laser. Two seed exposure times were used: 3 × 3 s and 3 × 9 s. 

The second step of stimulation was carried out in field conditions and involved the biostimulation of seedlings after the seeds had germinated in the field (K3 and K4). As in the case of the seeds, the biostimulation of young seedlings (irradiation of apical meristems) with red diode laser light was performed in two variants: 3 × 3 s and 3 × 9 s. Stimulation of plants in field conditions was carried out twice with a one-week interval using a modular platform.

Before harvest, 70 plants per each treatment were collected in order to determine selected morphological parameters, e.g., plant height, number of lateral shoots, plant weight, height of the first pod-setting, number of pods and seed number per pod. Seed harvest was carried out in the full maturity stage of soybean. After harvest, seed moisture was determined with the oven-drying method. In the paper, the seed yield is given calculated for normative moisture of 14%. The content of total protein and crude fat was determined using near infrared reflectance spectroscopy (NIRS) with InfraXact™ analyzer (Foss^®^) (Nils Foss Allé 1 DK-3400 Hilleroed Denmark). The content of nitrogen-free extract was calculated from the difference. The yield of total protein and crude fat was calculated based on the dry weight of seed yield and the content of seed components. The obtained data were subjected to statistical analysis using Statistica^®^ package (data analysis software system TIBCO software Inc., Palo Alto, CA, USA), version 13.1, while significant differences were evaluated with Tukey’s test using the significance level of *p* = 0.05

### 2.2. Economic Evaluation of Soybean Production 

Gross margin analysis was used to better understand the relationship between sales revenue and cost structures and assess the profitability of soybean production to enable informed decision making [19,20]. The gross margin analysis of soybean used to estimate the costs, returns, profitability or loss per hectare is given by the following relationships: GM = TR-TVC, where GM is gross margin, TR is total revenue and TVC is total variable cost [20]. The total revenue represents the value of the output from the farm (yield range from 1.52 to 3.22 t ha^−1^) multiplied by the prevailing market price (average EUR 489 ton^−1^). Total variable cost is a specific cost that varies directly with the level of production and includes expenditure on seeds, fuel and photostimulation application. Regarding the specific environmental conditions of soybean cultivation (optimal conditions), the economic analysis omits the cost of fertilization and pesticide applications. 

## 3. Results

The study compared different variants of biostimulation of seeds and plants, separately and in combination (Figure 1), in order to determine positive or negative effects as well as interactions, i.e., the synergistic effect of pre-emergence stimulation of seeds and post-emergence stimulation of plants. The statistical analysis showed significant differences in some of the morphological features of soybean (Figure 2 and Figure 3). Biostimulation of seeds in the K2 (3 × 9 S) and K6 (3 × 9 SP) treatments was shown to significantly affect plant height and the height of the first pod-setting above the ground surface. Taller plants were noted in the K2 (3 × 9 S) treatment and slightly shorter plants in the K5 plots (3 × 3 SP) in comparison with the control. The exposure time in the seed stimulation was shown to affect the formation of the first pods. Seed stimulation with or without plant stimulation increased the height of the first pod-setting (on average by 3–5 cm) in comparison to plants stimulated in field conditions. Both the number of plants per m^2^ and the shoot number per plant were slightly dependent on the stimulation treatment. 

The seed and plant biostimulation treatment was not shown to statistically significantly affect yield or yield structure (Figure 3 and Figure 4). However, it is worth noting the slight increase in seed yield and the values of individual yield structure parameters in treatment K4 (3 × 9 P) and the slight decrease in treatment K2 (3 × 9 S) in comparison with the control. The high seed yield in treatment K4 (3.22 t ha^−1^) clearly indicates the positive effect of post-emergence biostimulation of plants. In the K4 treatment, the plants had a higher pod number (28.6), pod weight (12.5 g), seed number per pod (2.24) and seed weight (6.72 g) than in the control. The yield was nearly as high (2.87 t ha^−1^) in the K2 treatment with pre-sowing stimulation (3 × 9 S), in which the 1000-seed weight (TSW) was much higher (131.4 g) than in the K4 treatment (101.3 g).

Biostimulation was shown to significantly affect the protein and fat content in the soybean seeds (Figure 5). Post-emergence biostimulation K4 (3 × 9 P) significantly increased the fat content in soybean seeds in comparison to the control, which translated to the value of the fat yield (599.4 kg ha^−1^). The integrated HUGO technology, involving pre-emergence biostimulation of seeds and post-emergence biostimulation of plants K5 (3 × 3 SP), significantly increased the protein content in the seed yield (331.7 g kg^−1^) in comparison to the control.

Biostimulation was not shown to significantly affect the efficiency of nitrogen uptake in the seeds (Figure 6). However, there was a slight increase in the nitrogen content in the K4 treatment (3 × 9 P) subjected to post-emergence stimulation (147.9 kg N ha^−1^). The protein and fat yield were not significantly affected by this experimental factor, but a significant increase in protein yield was noted in the plants subjected to post-emergence stimulation (924.6 kg ha^−1^) in comparison to the control (456.9 kg ha^−1^). 

The economic evaluation of soybean production depended on applying technology (K1–K6). The data collated in Figure 7 show that the amount of income depends on total revenue and total variable cost. The highest total revenue was noted in treatment K4, which was related with seed yield. The highest total variable cost was calculated for treatments K5 and K6 compared to the control. The value of gross margin was related to applying technology. The highest gross margin was obtained for treatment K4, whereas the lowest was obtained for K6 compared to the control. 

## 4. Discussion

The proposed innovative HUGO technology involving an integrated approach to biostimulation of plants shows a positive synergistic effect. Both the yield parameters and the quantitative parameters of the seed yield were significantly higher than in the control treatment. However, a stronger synergistic effect was observed for the quality parameters of the seed yield, in the case of both protein and fat content. The best qualitative effects were obtained following the use of combination K5 (3 × 3 SP). Due to the novelty of the technology, the results presented above can only partially be compared with results reported in the literature. Previous research on pre-emergence laser stimulation of seeds of various crop species has shown a positive effect of photostimulation on the yield and chemical composition of seeds. Podleśna et al. [21] analyzed the effect of pre-sowing biostimulation of seeds on the growth and development of pea (*Pisum sativum*) plants. The authors demonstrated that stimulation with laser light improved the morphological features of the plants by increasing their height and leaf area. Seed stimulation was shown to improve pea yield. The increase in seed yield was due to the higher pod number and seed number per plant, whereas no significant changes were observed in the seed number per pod. 

Pre-sowing biostimulation of seeds has also had a beneficial effect on the yield of other crop plants. In the case of oat (*Avena sativa*), the increase in the yield of irradiated plants relative to the control ranged from 16.5% to 23.2%, depending on the variety [22]. A study on the effectiveness of pre-sowing stimulation of alfalfa (*Medicago sativa*) and red clover (*Trifolium pratense*) seeds also showed an increase in the green and dry matter yield of the plants [23,24]. Podleśny [25], in a study on broad bean, observed an increase in yield from 5.1% to 9.4% in comparison with the control, depending on the number of times the seeds were irradiated prior to sowing. The positive effect of light stimulation on broad bean (*Vicia faba*) yield was also confirmed in later studies conducted by Podleśny [26].

Among oilseed plants, most research on the effect of pre-sowing irradiation of seeds has been conducted in rapeseed (*Brassica napus* L. var. napus). A study by Starzycki et al. [27] showed a beneficial effect of He-Ne laser light on the resistance of rapeseed plants to blackleg. Research on the content of nutrients in rapeseeds showed no significant differences following pre-sowing application of red light [28]. Makarska et al. [29] analyzed the effect of pre-sowing photostimulation on seed quality and showed that the use of a He-Ne laser did not alter the protein content in wheat grain. A study analyzing the effect of irradiation on maize (*Zea mays*) seeds showed higher yield resulting from increased grain number and weight [30].

## 5. Conclusions

Although numerous studies have been carried out in controlled conditions, there have been no field experiments on the photostimulation of soybean. Innovative research was carried out to assess the usefulness of complementary photostimulation of seeds and plants for the optimization of agricultural practices for soybean. We showed that pre-sowing photostimulation of seeds mainly has a positive effect on 1000-seed weight, whereas post-emergence photostimulation of plants positively affects other parameters of the yield structure, i.e., pod number, pod weight, seed number and seed weight, resulting in increased yield. The use of complementary photostimulation of both seeds and plants in field conditions led to a significant increase in protein and fat content in the soybeans. Further studies are needed on the complementary photostimulation of seeds and plants, since tested combinations of laser irradiation ought to also include yield quantity.

## Figures and Tables

**Figure 1 plants-11-01649-f001:**
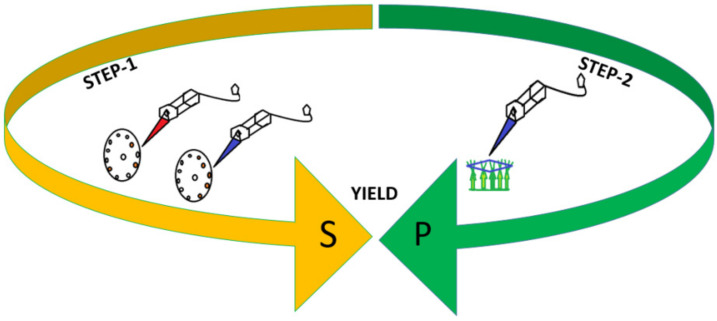
Complementary plant biostimulation. Positive synergistic effect of combining pre-emergence biostimulation of seeds with post-emergence biostimulation of plants.

**Figure 2 plants-11-01649-f002:**
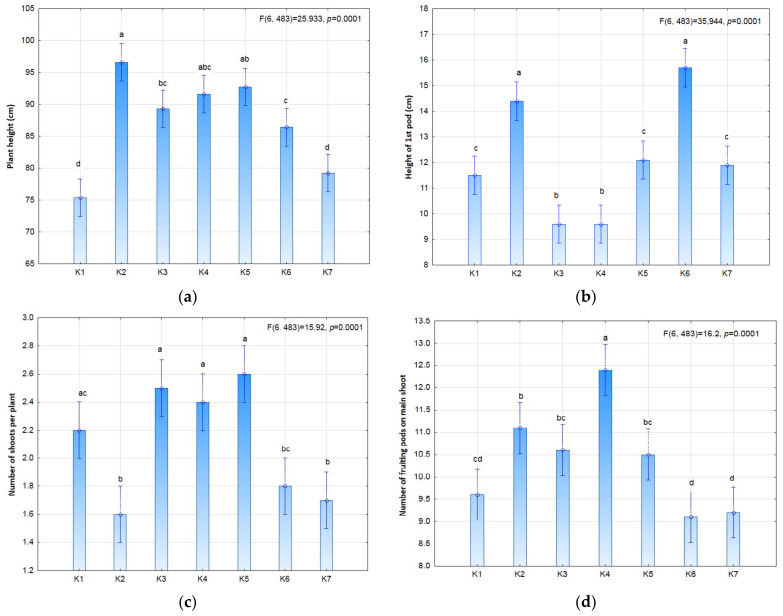
Selected morphological features of soybean (**a**) plant height; (**b**) height of first pod; (**c**) number of shoots per plant; (**d**) number of fruiting pods on main shoot) depending on photostimulation treatment. Different letters indicate significant differences between means. Significant effects at *p* < 0.05.

**Figure 3 plants-11-01649-f003:**
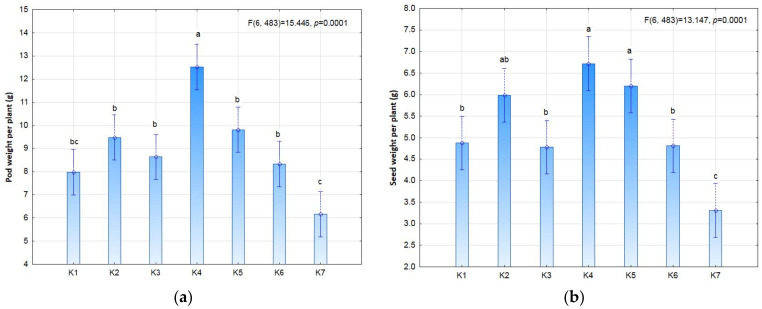
Selected morphological features of soybean (**a**) pod weight per plant; (**b**) seed weight per plant; (**c**) stem weight per plant; (**d**) dry weight of plant) depending on photostimulation treatment. Different letters indicate significant differences between means. Significant effects at *p* < 0.05.

**Figure 4 plants-11-01649-f004:**
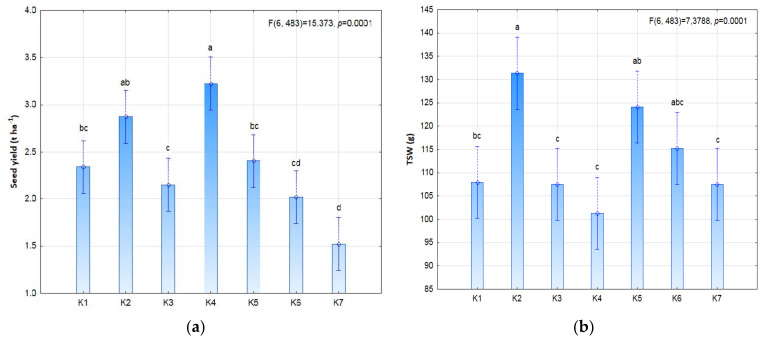
Seed yield (**a**), and yield parameters (**b**) TSW; (**c**) number of pods per plant; (**d**) seed number per pod) of soybean depending on photostimulation treatment. Different letters indicate significant differences between means. Significant effects at *p* < 0.05.

**Figure 5 plants-11-01649-f005:**
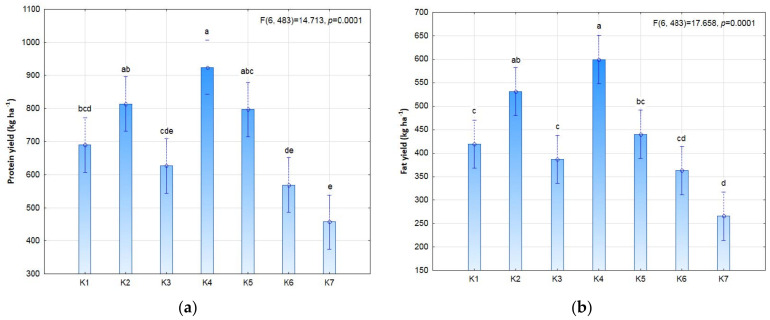
Protein yield (**a**) and fat yield (**b**) of soybean depending on photostimulation treatment. Different letters indicate significant differences between means. Significant effects at *p* < 0.05.

**Figure 6 plants-11-01649-f006:**
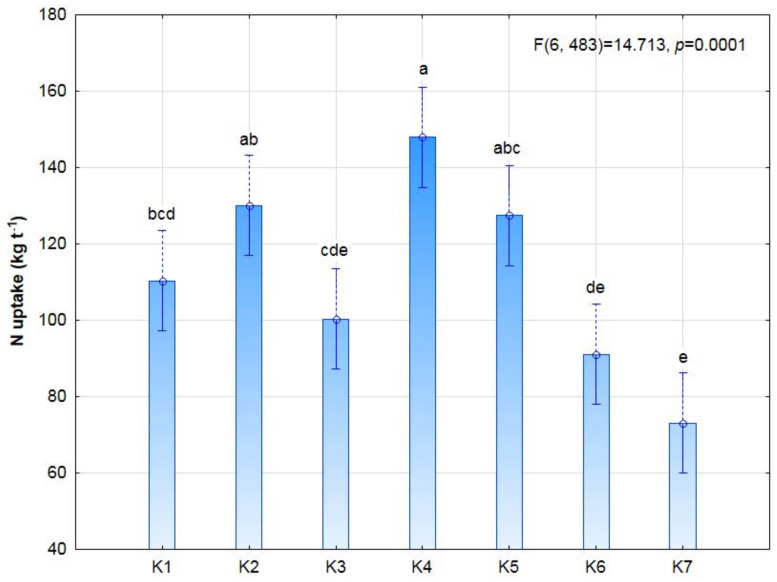
Nitrogen uptake with soybean yield, depending on photostimulation treatment. Different letters indicate significant differences between means. Significant effects at *p* < 0.05.

**Figure 7 plants-11-01649-f007:**
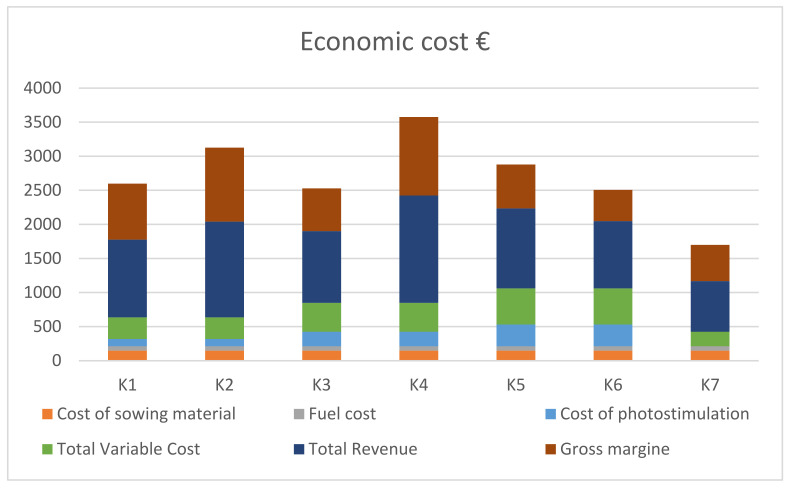
Average gross margin and variable costs (EUR ha^−1^ year^−1^) of soybean with (K1–K6) and without (K7) photostimulation.

**Table 1 plants-11-01649-t001:** Experimental design.

Combination	Seed Stimulation before Sowing (S)	Plant Stimulation (P)
**K1**	3 × 3	-
**K2**	3 × 9	-
**K3**	-	3 × 3
**K4**	-	3 × 9
**K5**	3 × 3	3 × 3
**K6**	3 × 9	3 × 9
**K7**	Control—no stimulation	

## Data Availability

Not applicable.

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
