# Peer review of "Complementary Photostimulation of Seeds and Plants as an Effective Tool for Increasing Crop Productivity and Quality in Light of New Challenges Facing Agriculture in the 21st Century—A Case Study"

_plants, 2022, doi:10.3390/plants11131649_

Round 1
Reviewer 1 Report
The paper is a preliminary study on the influence of photosimulation on a soybean variety. The results obtained are still few to allow generalizations. Further studies on improving the nutrient content of crop plants by photostimulation are needed.
Regarding the experiment performed for this paper, I would have some questions:
How many individuals from each variant did you analyze?
The size of the lot is specified - 7 sqm and the distance between the rows (25 cm) - but how many individuals were actually analyzed in each variant? If the plant density is 40-49 / sqm, does it imply that 280-343 individuals / sample were analyzed?
What is the origin of plant seeds? Has any preliminary study been conducted on seed germination?
Can the method be applied to large cultivated areas?
Can you appreciate the economic efficiency of such a method? Can the production surplus compensate the expenses for the placement of the installations in the cultivated field?
Author Response
Dear Reviewer,
We would like to thank for all comments and sugestions, which were included in corrected manuscript.
We included information about individual plants. We also inluced information from were we bougth high quality seeds material.
Preliminary studies dedicated to seeds germination were conducated, and proved that we bought high quality seeds from seeds company.
We also did very simple economic calculation as a respons for your suggestions, however, it was not a case of the study.
Thank you very much for support. The attached manuscript includes corrections.
Kind regards,
Authors

Reviewer 2 Report
The authors describe the treatment of soybean seeds and plants with laser photostimulation and its effect on crop productivity and quality.
Please find suggestions for improving the manuscript below.
I would suggest transferring the text which describes the concept of HUGO technology from the materials and methods part into the introduction.
Please describe in more detail in the introduction what is known about the effect of laser photostimulation on the physiology of plants. Why does this treatment increase crop productivity and quality? Are different effects expected when either seeds or young plants are treated with laser? What are the differences here? What was the rational behind combining both treatments? Please give more information.
In the introduction several studies are cited that showed the positive effect of laser photostimulation on different plants. Please state here if data are available also for soybean and if so, describe what was found in detail.
The chemical composition of the seeds by near-infrared spectroscopy (NIRS) should be described in more detail in the materials and methods part, so that the experiments can be repeated.
Please give the number of plants per treatment that were analyzed in the materials and methods part.
I would highly suggest presenting the data that are shown in tables in form of diagrams. This is much easier to grasp for the reader. Please add the number of plants (n) and error bars, so that the reader gets an impression of the reliability of the data set in addition to the statistical tests.
Please add scientific names for all plants that are mentioned.
Author Response
Dear Reviewer,
I would like to thank for all comments and suggestions, which were included in corrected manuscript.
We did correction in introduction section based on your suggestions and we inserted information about effect of photostimulation on the physiology of plants as well information why we observed increase of crop productivity after photostimulation. We also described in details some outcomes, specially for soybean. However, all references are included.
The methodology sections was corrected. We added information about plant density, and which tool was used to analyze yield quality. We did not put specific methodology for NIRS, since the analysis is not specific. We also changed the form of outcome presentation. The table were replaced by figures. For english names of species we added latin names in bracket. However we did it every first time mentioned.
Thank you very much for support. The attached manuscript includes corrections.
Kind regards,
Authors

Round 2
Reviewer 2 Report
All points raised were address satisfactorily.